# Potential Role of Soluble Toll-like Receptors 2 and 4 as Therapeutic Agents in Stroke and Brain Hemorrhage

**DOI:** 10.3390/ijms22189977

**Published:** 2021-09-15

**Authors:** Josh Lua, Kanishka Ekanayake, Madison Fangman, Sylvain Doré

**Affiliations:** 1Department of Anesthesiology, College of Medicine, University of Florida, Gainesville, FL 32610, USA; joshlua@ufl.edu (J.L.); kekanayake@ufl.edu (K.E.); maddiefang@ufl.edu (M.F.); 2Center for Translational Research in Neurodegenerative Disease, Departments of Psychiatry, Pharmaceutics and Neuroscience, McKnight Brain Institute, University of Florida, Gainesville, FL 32610, USA

**Keywords:** bleeding, decoy, hemin, inflammation, methemoglobin, shedding, therapy

## Abstract

Hemolysis is a physiological condition in which red blood cells (RBCs) lyse, releasing their contents into the extracellular environment. Hemolysis can be a manifestation of several diseases and conditions, such as sickle cell disease, hemorrhagic stroke, and trauma. Heme and hemoglobin are among the unique contents of RBCs that are released into the environment. Although these contents can cause oxidative stress, especially when oxidized in the extracellular environment, they can also initiate a proinflammatory response because they bind to receptors such as the Toll-like receptor (TLR) family. This review seeks to clarify the mechanism by which TLRs initiate a proinflammatory response to heme, hemoglobin, and their oxidized derivatives, as well as the possibility of using soluble TLRs (sTLRs) as therapeutic agents. Furthermore, this review explores the possibility of using sTLRs in hemorrhagic disorders in which mitigating inflammation is essential for clinical outcomes, including hemorrhagic stroke and its subtypes, intracerebral hemorrhage (ICH), and subarachnoid hemorrhage (SAH).

## 1. Introduction

Toll-like receptors (TLRs) are a class of membrane-bound proteins used to recognize external stimuli such as molecular patterns or gradients, and they initiate an immune response as part of the human innate immune system. The toll gene was originally discovered in Drosophila embryos [1]. It was later discovered that homologous forms in plants, insects, and vertebrates detect pathogen-associated molecular patterns (PAMPs) and damage-associated molecular patterns (DAMPs) and can catalyze an immune response through two inflammatory cascades: the Myeloid differentiation primary response 88 (MyD88)-dependent pathway and the TIR-domain-containing adapter-inducing interferon-β (TRIF)-dependent pathway [2]. Currently, there are 10 identified TLRs in humans (1–10) and 12 in mice (1–9, 11, 12, 13), with structurally similar TLRs 1–9 present in both [3]. However, mouse TLR10 is not functional due to a retrovirus insertion, and TLRs 11, 12, and 13 have disappeared from the human genome. TLRs are further categorized into two subcategories, cell surface TLRs and intracellular TLRs. TLRs 1, 2, 4, 5, 6, and 10 are expressed on the plasma membrane, whereas TLRs 3, 7, 8, 9, 11, 12, and 13 are expressed in the endosome, endoplasmic reticulum, lysosomes, and endolysosomes. Surface TLRs, such as TLR4, are widely recognized to play a key role in innate immunity. They are capable of recognizing endotoxins such as lipopolysaccharides (LPS), as well as other components of microbial membranes, including lipids and proteins [4]. On the other hand, intracellular TLRs are capable of recognizing bacterial and viral nucleic acids [5]. However, more recent research has elucidated the importance of TLRs and innate immunity regarding DAMPs, notably heme [4].

Hemolysis constitutes the breakdown of red blood cells (RBCs), resulting in the release of hemoglobin (Hb) into the extracellular environment. Hemoglobin can undergo a variety of biochemical transformations, including oxidation to methemoglobin (metHb) or breakdown to its component heme [6]. Hemolysis is associated with a wide variety of diseases and pathophysiological states, such as sickle cell disease [7], malaria [8], hemorrhagic stroke [9], and trauma. In all of these diseases, hemin, the oxidized form of heme, is a DAMP that triggers inflammation. Recent literature also reports that metHb and its further oxidized form, ferryl hemoglobin (FHb), trigger inflammation. This paper reviews the data that DAMPs are endogenous ligands of TLR2/4, triggering inflammation in hemorrhagic disorders such as intracerebral hemorrhage (ICH) and subarachnoid hemorrhage (SAH).

Additionally, there are free-floating variants of TLRs known as soluble TLRs (sTLRs). These free-floating protein complexes are considered structurally identical to their membrane-bound counterparts but do not participate in the TLR pathway. Instead, they reduce inflammatory responses by competing with TLRs for ligands. Langjahr et al. showed that sTLR2 is formed by post-translational modification of full-length membrane-bound receptors [10]. Specifically, sTLR2 production has been positively correlated with A disintegrin and metalloproteinase domain-containing protein 10 and 17 (ADAM10/ADAM17) activation and negatively correlated with TLR2 juxtamembrane region knockouts. The mechanism by which this process is regulated remains unknown. A murine model study conducted by Iwami et al. in 2000 was among the first to demonstrate an alternate splice variant of murine TLR4 (mTLR4) that results in naturally occurring soluble mTLR4 (smTLR4), although further studies have not been conducted on this alternative splicing nor what triggers it [11]. In vitro, LPS stimulation of human endothelial cells has been shown to induce TLR4 ectodomain shedding in a p38 mitogen-activated protein kinase (MAPK)/ADAM17-mediated pathway [12]. It is unknown whether these ectodomains are the same as sTLR4s circulating in humans, although the study found that the ectodomains were not taken up by endocytosis and that they had a molecular mass of ~48 kDa, much smaller than the known molecular weight of TLR4 of ~95 kDa. Similar to its membrane-bound counterpart, sTLR4 has been shown to interact with myeloid differentiation factor 2 (MD2) to bind LPS, thereby acting as a decoy receptor for its membrane-bound counterpart [13].

Current literature supports the expression of sTLR2 and sTLR4 in the blood [14,15,16], as well as in the mucosa, as evidenced by their presence in saliva [17,18,19]. Zunt et al. analyzed precleaned saliva samples with anti-human TLR4 antibodies and found polypeptides of 90, 78, 54, and 44 kDa [17]. The 54-kDa signal was the weakest. They also found that sTLR4s could inhibit tumor necrosis factor-α (TNFα) production in human macrophages stimulated with LPS. AlQallaf et al. found similar results, including the potential active sequestration of microbial ligands by sTLR4 [19]. 

## 2. Characterization of TLR Pathways

To initiate a signal transduction event, membrane-bound TLR4 receptors must first bind with extracellular myeloid differentiation factor 2 (MD2) and form the TLR4-MD2 complex [2]. Agonist activation causes this complex to recruit another TLR4-MD2 complex and undergo homodimerization. Specific agonists may require helper molecules to successfully bind to the complex. For example, LPS must first be bound to LPS-binding protein (LPB), found in plasma, and membrane-bound CD14, which delivers LPS-LPB to TLR4-MD2 [3]. Subsequently, the cytosolic toll receptor identity region (TIR) domain will symmetrically associate and create the signaling platform for adaptor recruitment. Unlike TLR4, TLR2 must first form a heterodimer with TLR1, TLR4, or TLR6 before it interacts with a ligand [20]. Homodimerization of TLR2 has been proposed, but induced homodimerization of human and mouse TLR2 hybrids hT2V9 and mT2V6 have failed [21]. This result does not indicate that TLR2 homodimerization is impossible but rather that our current observational techniques have limits. Hereinafter, the activation signal can diverge along two pathways, MyD88 or TRIF. Notably, all TLRs, except for TLR3, use the MyD88 pathway, whereas TLR3 uses the TRIF pathway. Additionally, TLR4 is capable of using both pathways, whereas TLR2 can use only the MyD88 pathway (See Figure 1).

### 2.1. MyD88-Dependent Pathway

Upon initial stimulation, the cytoplasmic portion of TLRs such as TLR2 binds to the MyD88 adaptor protein, which recruits IL-1 receptor-associated kinase (IRAK) 4, IRAK-1, IRAK-2, and IRAK-M [3]. Although it is possible to continue the pathway unassisted, TLR4 may receive assistance from MyD88 adaptor-like protein (MAL) [22]. MAL, also known as TIRAP, recruits the MyD88 adaptor protein to TLRs within the plasma membrane. After the IRAK family of protein kinases are activated, a phosphorylation/ubiquitination cascade is triggered, beginning at the E3 ligase TNF receptor-associated factor 6 (TRAF6) [3]. TRAF6 catalyzes the synthesis of Lys63 (K63) polyubiquitin chains on IRAK-1 in conjunction with the dimeric E2 ubiquitin-conjugating enzymes Ubc13 and Uev1A. The regulatory components of mitogen-activated protein kinase kinase kinase 7, also known as TAK1, then bind to the polyubiquitin chains on IRAK-1 and successfully activate, which allows it to phosphorylate MAPKs Erk1, Erk2, p38, and Jnk. These proteins activate the various transcription factors, including activator protein-1 (AP-1), which is responsible for activating the nuclear factor-κB (NFκB). NFκB concludes the cascade by allowing the expression of genes necessary for inflammatory responses.

### 2.2. TRIF-Dependent Pathway

The TRIF-dependent pathway, also known as the MyD88-independent pathway, similarly relies on many of the mechanisms found in the MyD88 pathway. While both TLR3 and 4 are capable of signaling through this pathway, TLR4 requires the TRIF-related adaptor molecule (TRAM) to recruit TRIF, whereas TLR3 can signal directly through TRIF [23]. Unlike the MyD88 TLR4 adapter, TIRAP, TRAM must initially dissociate with TLR4 after activation to continue TRIF signaling [24]. Through a ubiquitination-dependent mechanism using the IRAK family of proteins, the TRIF adaptor recruits TRAF6 and receptor-interacting protein 1 (RIP1), which can then activate TAK1 [3]. After this activation, the cascade functions in an identical manner to that of the MyD88 pathway.

### 2.3. Complications of the Signaling Pathway

Additional complications often occur in both the TLR2/4-mediated pathways, as well as other independently mediated pathways, which can vary the inflammatory response or completely alter it. These outcomes can arise through the multitude of intermediates in the signaling pathway, as well as the various types of cells/locations in which TLR is expressed. Thus, in diseases like ICH and SAH, even minor complications to an inflammatory pathway, such as the TLR4/NFκB pathway, can affect clinical outcomes [25].

Evidence suggests that enzymes such as heme oxygenase-1 (HO1) are capable of downregulating the inflammatory response commonly seen in TLR activation [26]. Huang et al. demonstrated decreased expression levels of TLR2, TLR4, IRAK1, and TRAF6 in liver tissue treated with cobalt protoporphyrin, an HO1 inducer, following ischemia-reperfusion injury. Although the exact mechanism remains unknown, the anti-inflammatory properties of HO1 most likely stem from its degradation of heme to carbon monoxide, ferrous iron, and biliverdin. This has the two-fold effect of removing heme, a putative TLR ligand [27], and adding CO, a known anti-inflammatory gasotransmitter [28,29], and adding biliverdin and bilirubin, known to antioxidants. Regardless of the cause, their results signify a reduced association between MyD88 and TLR2/4, as well as IRAK1 and TRAF6, which would, in turn, reduce the expression of further downstream signaling. Furthermore, phosphorylation of NFκB was also reduced, which suggests an interaction between HO1 and the TLR2/4 Myd88 pathway. 

Rnf112, though unrelated to HO1, also appears to downregulate TLR4 and the inflammatory pathway, at least in the context of ICH. Zhang and Zhang found that endogenous expression of Rnf112 was decreased following ICH [30]. Rnf112−/− mice had increased expression of the TLR4/NFκB pathway, a finding that was reflected by considerably worse anatomical and functional outcomes, measured through brain water content and neurological deficit score (NDS), respectively. Despite being poorly characterized in current literature, the mechanism of Rnf112 appears to be separate from the mechanism of heme and HO1, as the protective effect was still found in LPS treated cells in vitro.

Receptors and cytokines are not the only types of proteins that can affect TLR2/4. For example, Tenascin-C is a protein present in the extracellular matrix that has been shown to affect anatomical outcomes such as blood–brain barrier permeability during SAH in murine models [31]. In addition, a more recent study by Liu et al. demonstrated that tenascin-C−/− mice had attenuated expression levels for proteins of the TLR4/NFκB pathway compared to their WT counterparts [32]. However, any therapeutic potential in tenascin-C must be qualified with the fact that the protein plays a key role in the differentiation of neurons and other cells [33]. Therefore, the possibility of long-term effects regarding this differentiation must be ruled out before moving forward.

TLR activation has also been shown to result in different responses depending on the cell type in which it is located. The presence of TLR4 is required for macrophages to polarize toward the M1 proinflammatory phenotype in the presence of heme [34]. The WT SV129 mice challenged with heme and the TLR4 inhibitor TAK-242 showed decreased expression of hepatic and splenic macrophage M1 markers MHCII and CD86. Hepatic macrophages exposed to TAK-242 also expressed diminished heme-driven induction of IL-6 and TNFα. 

For the remainder of the review, the primary focus will be TLRs 2 and 4 due to the extent of research conducted regarding their proinflammatory response to DAMPs, as well as their sTLR counterparts. Gaps remain in the research regarding the exact molecular mechanism by which TLRs regulate intracellular processes, although the receptor appears to be pleiotropic for different cell types. For example, TLRs, such as TLR4, have been shown to play a role in the N1/N2 polarization of neutrophils after stroke [35], whereas TLRs 1, 2, 3, and 4 have been shown to regulate heme and iron export in macrophages [36]. Both mechanisms play a key role in the functioning of these cells; however, the location at which the signaling cascades diverge from the already studied inflammatory pathway is unknown. Furthermore, the molecular mechanisms of soluble TLR formation and their interactions with PAMPs and DAMPs are poorly characterized. Both gaps in the literature must be addressed before sTLR2/4 can be considered for clinical applications.

## 3. Hemoglobin and Its Derivatives Interact with TLRs to Produce an Inflammatory Response in Multiple Cell Types

RBCs can be lysed in a variety of diseases, either in systematic events such as sickle cell disease and septic shock or in more localized events such as trauma, SAH, and ICH. Although heme, hemin, and hemoglobin have demonstrated a synergetic effect in the inflammatory response produced by TLR ligands such as LPS [37], there is also substantial interest in finding which of these DAMPs are responsible for the inflammatory response because this could help guide future research for hemorrhagic events and acute central nervous system diseases such as SAH and ICH. Current research has electronically simulated two potentially viable heme activation sites on TLR4/MD2 that are distinct from the LPS binding site [38]. Further analysis on HEK cells using NFκB reporter assays showed increased activity in heme, LPS, and heme+LPS (*p* < 0.01 for each), but this increase was attenuated if TLR4, MD2, or CD14 was absent. That finding suggests that each protein is crucial to the activation of TLR4. Notably, Belcher et al. did not find a significant increase in the heme+LPS groups compared to the heme or LPS groups alone. This finding corroborates an earlier study by Figueiredo et al., which used MTS510, an anti-TLR4/MD2 antibody that inhibits LPS activation of the receptor, to show that heme is a ligand of TLR4 in murine macrophages, albeit at a different site [27]. Finding and inhibiting the binding site of heme/hemin to TLR4 should be studied considering the role their interaction plays in hemorrhagic stroke, which has been well characterized in a previous review by Fang et al. (See Figure 2) [39].

However, a DAMP can serve as a sensitizer, which means it increases the response to a ligand while not functioning as a ligand itself [40]. In this regard, in vitro studies may have limitations and advantages for studying hemoglobin. They are not subject to the variety of physiological reactions involved in the oxidation and degradation of hemoglobin in the extracellular environment [6]; conversely, they can reveal that it is not the hemoglobin itself but one of its derivatives that serves as the primary TLR ligand [41]. Furthermore, in vitro models may show that a single cell type does not provide the main response to a ligand, but they cannot show the overall biological response that arises in an in vivo model. Many studies have opted to use both to strengthen their body of evidence.

One consideration when studying TLR ligands is the possibility of contamination with their PAMP counterparts. These PAMP ligands include LPS and, in the case of TLR2 specifically, BLP and LTA [42]. In a review of 23 investigative TLR agonists, eight failed to produce results in subsequent studies that used further LPS purification methods [40], illustrating the need for a methodology that avoids contamination. Common strategies for preventing contamination include using polymyxin B [41,43], an antibiotic with the ability to bind and sequester LPS, as well as LPS-removal column chromatography. In addition, other methods can be used to assure researchers of contaminant exclusion, such as detecting LPS with a Limulus amebocyte lysate assay [44] or inactivating a sample of the protein under investigation and adding it to culture [41]. Thus, an important distinction must be drawn between being a sensitizer for other ligands and being a ligand as a compound that is only a sensitizer with less therapeutic potential in events involving internal hemorrhage and “sterile” inflammation ICH and SAH.

### 3.1. Microglia and Other Glial Cells

Evidence suggests that during hemorrhagic events in the brain such as SAH and ICH, TLR activation can be divided into an early period, mediated by the TLR4/MyD88 pathway, and a late period, mediated by the TLR4/TRIF pathway [45]. Researcher KA Hanafy used C57BL/6 mice to determine that the early period usually occurs within 5 days of SAH and is primarily mediated by microglia, most likely resulting in early brain injury (EBI) after the onset of the disease. According to Hanafy, however, the late period begins 7 days after SAH and is no longer dependent on microglia [45]. The exact mechanism that causes TLR4 to switch pathways in this instance is unknown; however, after further testing using an in vitro model, Hanafy proposed several theories, including time-dependent activation of adaptor molecules, the functional inability of the mouse model to differentiate between pathways, and influences by extra chemical and cellular factors in the brain. This shift is thought to be represented biologically by changing the TLR4/MyD88 pathway to the TLR4/TRIF pathway. EBI is of interest to researchers because it is strongly predictive of patient outcomes and often leads to delayed cerebral ischemia (DCI) [46]. A preclinical model of SAH in C57BL/6J mice found that functional outcomes were strongly connected with levels of TLR4 expression and microglial phenotypes, which they controlled by using the experimental drug curcumin [47]. Of the DAMPs of interest during SAH and other hemorrhagic events, it appears that heme/hemin [48] and metHb [44] act as endogenous ligands of TLR4, although the latter appears to be debatable. 

### 3.2. Neurons

Although current literature does not implicate neurons as a major source of TLR expression, Zhang et al. found that the levels of neuronal cell death were directly related to the levels of overall TLR4 expression [49]. Combined with the finding that TLR4 is mainly expressed in microglia, this finding implies that neuronal cell death during SAH is mostly connected to microglia and the TLR4/MyD88 pathway within them. By contrast, another study by Chen et al. found that the relationship between microglia and neurons may be reciprocal. The reason is that the inhibition of RNF216 mRNA expression in rat neurons, which is responsible for the ubiquitination and degradation of TLRs, has been connected to the amelioration of brain injury and better functional outcomes after SAH [50]. However, these results may have been influenced by the fact that the expression of RNF216 also reduces the expression of AMPAR GluR2. This results in excessive neuronal Ca^2+^ permeability, eventually leading to cell death and dysfunction after SAH. Therefore, more studies are needed on the relationship between neurons, astrocytes, and microglial cells during hemorrhagic events such as SAH and ICH because it is becoming increasingly clear that microglia are not the sole mediator of inflammation and neurotoxicity.

### 3.3. Macrophages

Macrophages, like microglia, play an important role in localized injuries such as SAH. Additionally, unlike microglia, macrophages can play a significant role in systemic inflammatory regulation because of their prevalence throughout the body. Figueiredo et al. were among the first to demonstrate that heme is a ligand of TLR4 in macrophages and, in general, to use heme analog protoporphyrin IX, MTS510, and TLR4^−/−^ mice to support the concept of the TLR4/MD2/MyD88-mediated inflammatory pathway [27]. However, other early data suggested that hemoglobin and its oxidized counterparts, metHb, and FHb, did not affect the inflammatory pathway in bone-marrow-derived macrophages in vitro, as measured by TNF and IL-6 expression [41]. The contradictory result could be because Silva et al. conducted the experiment in vitro. However, it should be noted that the in vivo portion of the experiment found that FHb, but not metHb or Hb, had inflammatory effects when injected intraperitoneally. Despite these findings, more recent evidence suggests that macrophages use the TLR4 pathway during hemolytic events. Evidence of this use was seen in a C57BL/6 murine model of ICH that demonstrated TLR2/TLR4 heterodimer formation in infiltrating macrophages. Homodimerization subsequently activated a pathway involving many notable inflammatory components, including IL-23, which further stimulated γδ T cell production of IL-17 [51]. This finding indicates that the TLR2/TLR4 heterodimer and the MyD88 pathway are responsible for the inflammatory response after ICH, which may be a significant cause of secondary injury. Meanwhile, in a systemic model of hemolysis in B6J and SV129 mice, Vinchi et al. found that cell-free hemin injected intravenously affected a variety of organs, including the liver and spleen [34]. However, the most relevant finding was hemin interaction with TLR4 in endothelial cells and BMDMs to produce inflammatory cytokines, which thereafter caused macrophages to transpose toward a proinflammatory M1 phenotype [34]. Nonetheless, the inflammatory cascade was reversed when the TLR4 receptors were neutralized, either by using TLR4^−/−^ mice (*p* < 0.05 vs. WT) or treating the WT mice with heme and TLR4 inhibitor TAK-242 (*p* < 0.05 vs. heme alone), as well as when the heme ligand was neutralized using its binding protein hemopexin (*p* < 0.05 vs. heme alone). Overall, these findings suggest that macrophages are activated toward inflammatory phenotypes when exposed to the products of hemolysis such as heme but not hemoglobin and its oxidized counterparts.

### 3.4. Neutrophils and Other Immune Cells

Neutrophils, such as macrophages, play an important role in inflammation and the immune response. Therefore, it is unsurprising that both myeloid cells express TLR2 and TLR4 [35,42,43]. The TLR4 receptor in neutrophils plays an inflammatory role in SAH similar to microglia, although it remains an independent process. Chimeric mice with microglia expressing TLR4 and neutrophils expressing non-TLR4 had neuroprotection during ischemic stroke, as modeled with permanent middle cerebral artery occlusion [35]. Regardless of the improved anatomical outcomes, a significantly higher number of neutrophils infiltrated the brain in the non-TLR4 expressed neutrophil mice than in their TLR4-expressing counterparts (*p* < 0.05). This is most likely connected to the increased expression of anti-inflammatory N2 phenotypes in non-TLR4 expressing mice (*p* < 0.05). TLR2 in neutrophils, on the other hand, tends to play a more important role in hemolytic Gram-negative infections. In those situations, the release of lipoteichoic acid (LTA) and metHb, which serve as PAMPs and DAMPs, respectively, activate TLR2 pathways in monocytes, T-cells, and neutrophils. Although TLR2 agonists, such as LTA, normally decrease neutrophil apoptosis, neutrophil apoptosis was elevated in high concentrations of metHb, suggesting that cell-free hemoglobin has cytotoxic characteristics [42]. 

### 3.5. Endothelial Cells

Although endothelial cells do not function as immune cells, they still play important roles in inflammatory complement systems. Due to the existence of endothelial cell cultures such as HUVECs and HMEC-1, many in vitro studies on the interaction of hemoglobin with TLR2/4 use endothelial cells. Lisk et al. found that hemoglobin, metHb, and FHb interact with endothelial cells in a MyD88-dependent but TLR4-independent pathway, as evidenced by the fact that the increased NFκB and HIF expression was not attenuated by TAK-242 treatment [52]. Interestingly, the expression of these markers was attenuated by superoxide dismutase and catalase treatment, which suggests that the creation of reactive oxygen species leads to cytotoxic effects and increased endothelial permeability after the release of hemoglobin into the bloodstream. An earlier study by Silva et al. found contradictory results in endothelial cells [41]. FHb, but not hemoglobin and metHb, had JNK- and p38-dependent proinflammatory effects in HUVEC, which was also mediated through actin polymerization. Interestingly, the study also found that metHb-induced recruitment of neutrophils in an in vivo model of rats was not TLR4 dependent. The divergent results in these studies may be because of a difference in the biomarkers measured, although further research must be conducted on NFκB activation in these cells. Despite the differences, both studies agreed that TLR4 is not the primary mediator of inflammatory responses to hemolysate in endothelial cells.

## 4. Alternative TLR Pathways

In addition to competing against their soluble counterparts, TLR2 and 4 also compete against fellow TLR proteins. TLR7, despite being another member of the TLR receptor family, appears to serve an opposite role during hemorrhagic events such as ICH. Specifically, heme acts as a ligand to TLR7, but rather than stimulating a MyD88-mediated inflammatory pathway, TLR7 activates the BTK-CRT-LRP1 pathway, resulting in hemopexin production [53]. Hemopexin plays a vital role in heme scavenging, combined with cell-free heme’s role in inflammation. As mentioned earlier, a decrease in heme concentration should lead to decreased activation of the TLR4/MD2 complex and attenuate inflammatory reactions. 

More notably, TLR3/4 can also induce an anti-inflammatory and anti-apoptotic response through the TRIF-dependent pathway. Rather than proceeding through the TRAF6 adaptor molecule, activated TRIF recruits TRAF3, which interacts with TRIF-binding kinase (TBK) and IKKi. These two kinases phosphorylate IRF3, allowing it to dimerize and initiate interferon-β synthesis, which terminates the cascade [3]. It is currently unknown what determines whether TRAF6 or TRAF3 is activated in the TRIF-dependent pathway. Given the conflicting outcomes, determining the factor that causes the pathways to diverge may provide potential therapeutic targets. 

## 5. The Role of sTLRs in Reducing Hemorrhagic Inflammatory Responses

Soluble TLRs (sTLRs) act as some of the first inhibitors of the TLR inflammatory response. These free-floating extracellular domains of TLR2/4 reduce the expression of inflammatory genes by binding to PAMPs and DAMPs before they can interact with membrane-bound TLRs [54]. Currently, blood mononuclear cells (BMCs) are the only putative source of sTLR2 and sTLR4 after exposure to microbial ligands [16]; however, aortic endothelial cells have been reported to cleave TLR4 ectodomains after LPS exposure as a form of negative feedback [12]. Additionally, Sokól et al. have hypothesized that a minor amount of membrane TLR is released after necrosis; membrane fragmentation can result in similar production (See Figure 3) [15]. They also demonstrated elevated sTLR2/4 levels in cerebrospinal fluid after SAH using data drawn from 18 patients with SAH and acute hydrocephalus. This result contrasts to plasma production of sTLR in response to PAMPs, which usually peaks 2–4 h after exposure to microbial ligands [16]. It is unknown what causes this delay; however, it may be because additional time is required for BMCs to migrate across the blood-brain barrier or because of a difference in DAMP- vs. PAMP-activated sTLR production. Given the role of sTLRs as the first line of TLR inhibitors, it would make sense that increased expression would have an anti-inflammatory effect in events with a deluge of TLR ligands, such as SAH. However, no correlation was found between sTLR expression and clinical outcome. 

## 6. TLRs as a Therapeutic Target

In terms of minimizing inflammatory responses via the TLR pathway, there are two main schools of thought: the minimization of the ligand and the inhibition of the receptor. Because researchers generally agree that heme (hemin) is an endogenous ligand of TLR2/4, they are highly interested in finding a method of neutralizing the protein in vivo. Cell-free heme is naturally degraded to carbon monoxide, iron, and bilirubin by HO1 [55]. Furthermore, as mentioned in the TLR7 pathway, hemopexin is the serum protein with the highest binding affinity for heme [53]. Hemoglobin, though not a ligand of TLR4, degrades to heme; therefore, the binding and degrading of the cell-free form should be of equal concern to researchers. During hemolytic events, hemoglobin levels are nearly undetectable because of the presence of haptoglobin, the second most abundant protein in the bloodstream, which binds hemoglobin with a high affinity. Although hemopexin and haptoglobin bind their respective cell-free proteins and are ubiquitous throughout the body, they are also large, glycosylated proteins, which poses a challenge for commercial use. Furthermore, they would need to be used together because of the presence of heme and hemoglobin as hemolysis products [56]. In this respect, the lack of specificity of sTLRs provides an advantage because they can be used to generally inhibit the binding of ligands to their membrane-bound counterparts. Another potential direction for using TLRs as therapeutic targets is the induction of heme- and hemoglobin-scavenging proteins through drugs, e.g., using HO1 inducer compounds; through gene therapy; or indirectly through targeting TLR7, Nrf2, and similar anti-inflammatory proteins [57]. Additional tools would be the use of adeno-associated viruses to overexpress the respective sTLRs, an avenue that our laboratory is actively pursuing. 

Until further studies find a feasible way to address the products of hemolysis, the next best option to combat inflammation and promote neuroprotection is to address the receptor itself. Although Nrf112 is a potential target for downregulating TLR4 [30], there are no known drugs that target the receptor. Furthermore, there is also no evidence that Nrf112 acts on TLR2. siRNA is a direct method of inhibiting specific genes’ expression, and there have been experimental applications of siRNA to knock down the expression of key proteins such as TLR4 [44] and genes such as NFκB [52]. On the other hand, miRNA can target multiple mRNA transcripts, thus making it more effective in silencing multiple genes. miRNA like miR-140-5p [58] and miR146a [59] are two examples that target TLR4 and its pathways. However, concerns have been raised about the knockdown efficiency, as well as the toxic effects of nonspecific binding by siRNA and miRNA [60]. Antibodies can provide a more specific means of inhibiting TLR2 and 4; indeed, MTS510, a monoclonal antibody, has already been used to inhibit TLR4 expression in ischemic [61] and hemorrhagic stroke [27,62] in preclinical models.

In terms of drugs, TAK-242 has been classically used as a TLR4 antagonist [34,52,63,64] and can reduce inflammasome production to almost undetectable levels. The drug was synthesized by Takeda Pharmaceuticals [65]. Matsunaga et al. demonstrated that it had the ability to selectively bind to TLR4 in an in vitro model of HEK293 cells that express TLR4, MD2, and CD14 [66]. Although the therapeutic possibilities of the drug are interesting given its potency, they are limited by its complexity and the difficulty of producing the drug. Rosiglitazone is another drug that has been found to attenuate TLR4 expression in vascular smooth muscle cells in response to oxyhemoglobin exposure [67]. Unlike TAK-242, the drug is currently used to treat type II diabetes, so the ability to be repurposed without changing any current production makes it a more economically feasible option for researchers to pursue. Although there are other drugs in development that inhibit TLR4, both TAK-242 and rosiglitazone have the advantage of the blood–brain barrier [68,69]. Curcumin, a natural compound found in the rhizomes of Curcuma longa, has also been found to reduce TLR4 expression (*p* < 0.01) and increase functional outcomes as measured by the NDS (*p* < 0.05) in a SAH model with C57BL/6J mice [47]. However, these significant differences were not found in the TLR4^−/−^ mice. Although significant, the difference remained modest compared to TAK-242, and the half-life of the drug is relatively short; the question of which drug to produce remains an economic one rather than a scientific one. Other naturally occurring TLR inhibitors that have demonstrated neuroprotection in hemorrhagic stroke include plant-based flavonoids like fisetin [70] and luteolin [71], as well as astaxanthin, a carotenoid [49] (See Appendix A). 

Another promising alternative therapeutic method for inhibiting TLR2/4 is introducing sTLRs to compete with their membrane-bound counterparts for heme derivatives. More analysis is needed on the exact composition of sTLRs, but assuming they resemble the complete ectodomains of TLR2/4, they should bind the same ligands as their membrane-bound form. In fact, Iwami et al. showed that sTLR4 could inhibit the binding of LPS to its cellular counterparts in a murine model. Future research should investigate whether it can do the same for heme, metHb, and its related counterparts in a human model [11]. Similar results have been found with sTLR2 and its membrane counterpart regarding the bacterial lipoprotein Pam3Cys [14]. However, it should be noted that the kinetics of in vivo concentrations of sTLR2 and sTLR4 differ: the concentrations of serum sTLR2 are much higher than sTLR4 [16]. This suggests that an unknown mechanism of endogenous regulation may interfere with the therapeutic applications of sTLRs.

## 7. Clinical Studies and Proteomics

TLR and heme interactions have been poorly studied in clinical models, and sTLR interactions with DAMPs have been studied even less. Furthermore, given their innate role in SAH edema, experimental interventions regarding TLR interactions are often challenging to conduct. However, recent research has begun to explore the use of TLR and sTLR levels as accurate diagnostic tools in detecting hemorrhagic disorders that are generally difficult to detect. These include cerebral aneurysm ruptures, acute aortic dissection (AAD), and DCI [15,25,72] (See Appendix A). 

Serum TLR levels have shown a strong correlation with the severity of DCI and AAD. For example, in a study of 30 patients with aneurysmal SAH, higher TLR4 expression in peripheral blood mononuclear cells (PBMCs) on day 1 after SAH was an independent predictor of poor neurological outcome after 3 months (*p* = 0.004) [25]. Similarly, a study of 88 patients diagnosed with AAD showed a positive correlation between TLR4 expression and matrix metalloproteinase 9 (*p* < 0.001), which is an extracellular matrix protease capable of degrading the aortic wall and causing AAD [72]. 

As previously mentioned, a study of 18 patients demonstrated increased levels of sTLR2 and sTLR4 in the cerebrospinal fluid 3 days after SAH [15]. The strongest correlation was seen on day 5 (*p* = 0.037). However, the exact physiological purpose for this increase in levels is unknown. Since TLR signaling usually peaks from 2 to 6 h after aneurysm rupture, the delayed increase in sTLR production would appear to have little to no effect on TLR signaling. This was further demonstrated by a low correlation between sTLR2/4 levels and measurements of inflammation, namely white blood cell count and C-reactive protein (CRP) levels. There was also no correlation between sTLR2/4 levels and patient outcome. However, further investigation is needed to determine whether this is simply due to a lack of anatomical relevance for sTLR2/4 in hemorrhagic injury or if the levels of production are correlated with the severity of the damage.

Due to the lack of investigation into the role of sTLRs in acute hemorrhagic disorders, we reviewed the proteomics of sTLRs in other diseases. In a study of multiple sclerosis, a chronic inflammatory central nervous system disease, levels of sTLR2 but not sTLR4 were higher in patients with multiple sclerosis than in controls [73]. However, the levels did not change between the relapse and remission stages of the disease, despite the appearance of symptoms in the former. In a multicenter observational study of patients with alcoholic hepatitis, sTLR2 levels were correlated with measures of liver function such as the Model for End-Stage Liver Disease score and aspartate transaminase expressions. More importantly, they were positively correlated with cytokines IL-8, IP10 (also known as CXCL10), and TGFα [74]. In a study of colorectal cancer, high levels of CRP, an inflammatory marker associated with recurrence and poor patient outcomes, were associated with low/undetectable serum TLR2 levels [75]. A negative correlation with inflammatory markers is surprising considering the key role that TLR2 plays in inflammation. However, the current methodology of using ELISA to measure TLR expression cannot distinguish between the membrane-bound form and its soluble counterpart, the latter of which is most likely the form in serum. The form of TLR2 found in serum is most likely the soluble form, which supports the concept that sTLR2 acts as a decoy receptor. Nevertheless, more research is needed to characterize whether sTLR2 indeed causes this inverse correlation or whether other factors play a role. Furthermore, additional research is required to explain why sTLR4 did not play a significant role in these diseases, considering its importance in other inflammatory pathways. When considering the proteomics of sTLR2, age is another important factor to consider; a study found that sTLR2 levels in the saliva decrease with age, although it is unknown whether this observation applies to serum sTLR2 levels [18].

## 8. Discussion

Despite the well-studied effects of TLR2/4 in brain ischemia and hemorrhagic stroke, the role of sTLRs in these conditions remains a gap in the literature. Theoretically, these proteins resemble the ectodomains of the membrane-bound counterparts and should bind to the same ligands, which has been reflected repeatedly in studies concerning LPS. However, there is little evidence that the same can be said about their DAMP counterparts, heme and metHb. It is also unknown whether other proteins, such as MD2, are necessary for TLRs to bind DAMPs. Therefore, future in vitro research should investigate whether sTLRs can act as decoy receptors for heme, hemoglobin, and derivatives. In vivo research should focus on whether or not sTLRs can have an effect on their own in a preclinical model of hemorrhagic stroke, especially considering the role the blood–brain barrier can have in pharmacokinetics and the passage of proteins [76].

When considering the possible therapeutic application of sTLR in hemorrhagic and hemolytic diseases, the affinity of the soluble receptor to its DAMP ligands must be considered. For example, hemopexin may be a better option for binding heme, but the lack of specificity to a single ligand for sTLRs could be an advantage in the eyes of researchers. Furthermore, the production of sTLRs can be done either indirectly, i.e., through the induction of metalloproteases such as ADAM10/ADAM17, or directly, by isolating the gene responsible for the expression of TLRs. In terms of the latter, the use of recombinant receptor protein ectodomains in medicine is not unheard of. Buetler et al. have patented the technology to create soluble TNF receptors [77]. This patent was later sold to Immunex (Seattle, WA) and used to develop the drug Etanercept, which is a treatment for inflammatory conditions. However, a limitation of the current literature on sTLRs is the lack of understanding of how several factors can alter their expressions, such as age and alternative splicing. Overall, future research should focus on determining the mechanisms through which TLRs and sTLRs are regulated in vivo.

The importance of adaptor proteins is another factor that cannot be understated, especially considering future directions of research for TLRs. For example, the mechanism of determining a proinflammatory versus anti-inflammatory response through adaptor proteins such as MyD88, TRIF, TRAF3, and TRAF6 must be further studied to identify possible therapeutic targets. Although it may be simpler to silence TLR receptors altogether, the lack of distinction as to which downstream pathways are silenced may have unintended consequences. For example, TLR4 has been implicated to have a role in the polarization of neutrophils and macrophages toward proinflammatory phenotypes [35,63]. This feature may be desirable in diseases such as stroke but possibly detrimental when fighting infections. 

## 9. Conclusions

In this review, we focused on the proinflammatory effects of TLR signaling in reaction to heme, hemoglobin, and its oxidized derivatives, as well as the methods of regulation by molecular mechanisms. The past decades of research have demonstrated that TLRs play a significant role in the proinflammatory response, but our review has elucidated several points of contention that require further study. Several potential therapeutic targets that may help reduce inflammatory or oxidative stress due to exposure to heme during stroke have been identified, mostly in the regulation of sTLR variants. Although not all studies demonstrated a clear-cut correlation between serum sTLR and an anti-inflammatory effect, the varied results indicate additional areas for further research as other factors may be at play. Thus, although decoy sTLRs remain a promising candidate for human use, these unknown variables must be addressed in preclinical models before they can be transitioned to clinical trials.

## Figures and Tables

**Figure 1 ijms-22-09977-f001:**
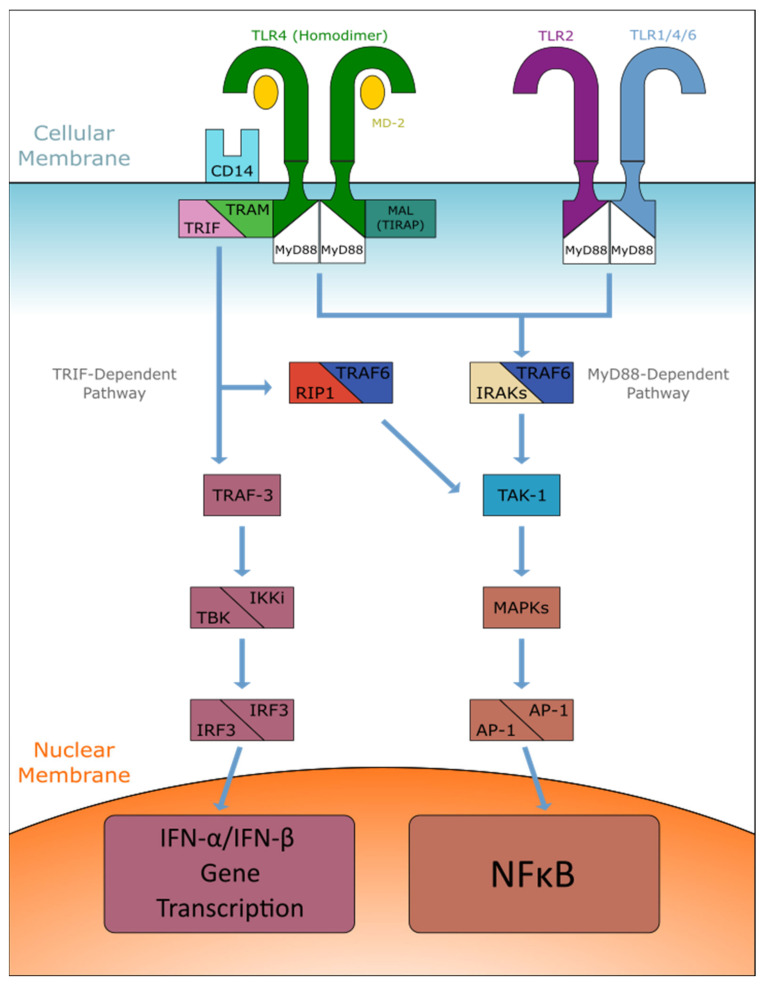
Illustration of the MyD88-dependent and TRIF-dependent (MyD88-independent) pathways. Current research supports the presence of both pathways when mediated by TLR4 homodimers and CD14, but only the MyD88 pathway when considering TLR2 heterodimers.

**Figure 2 ijms-22-09977-f002:**
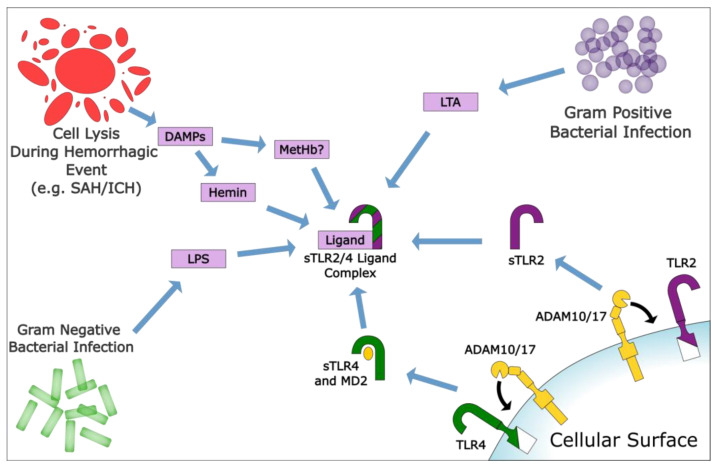
Illustration of the ADAM10/17 cleavage of membrane-bound TLRs to produce soluble TLRs (sTLRs) and the proposed binding of damage-associated molecular patterns (DAMPs) to sTLRs. Note that lipoteichoic acid (LTA) can only bind to TLR2, and the binding of methemoglobin (metHb) to TLRs is a matter of debate.

**Figure 3 ijms-22-09977-f003:**
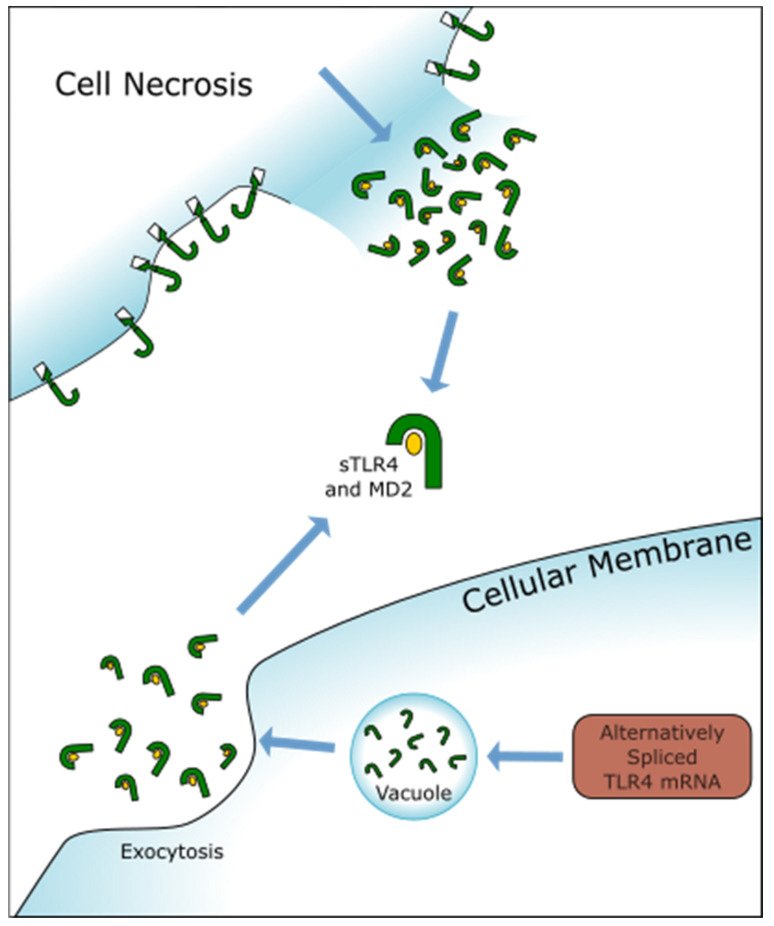
Illustration of the other two reported sources of sTLRs, cell necrosis, and alternatively spliced TLRs. It should be noted that necrosis is thought to only account for a small portion of the sTLR produced in humans, and alternatively spliced sTLR has only been observed in a murine model.

## Data Availability

The data presented in this study are available on request from the corresponding author.

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
