# Peer review of "Potential Role of Soluble Toll-like Receptors 2 and 4 as Therapeutic Agents in Stroke and Brain Hemorrhage"

_ijms, 2021, doi:10.3390/ijms22189977_

Round 1

Reviewer 1 Report

This work is detailed enough and satisfactorily covers the literature in this field.

I have only a minor comment: please, provide a sharper image for Fig 2.

Author Response

Reviewer 1

This work is detailed enough and satisfactorily covers the literature in this field.

I have only a minor comment: please, provide a sharper image for Fig 2.

REPLY: We edited the manuscript and have now included a high-resolution image for figure 2, with easier-to-read text.

Reviewer 2 Report

This systematic review by Josh Lua is a comprehensive and state-of-the-art presentation of the new insight into the potential role of soluble Toll-Like receptors 2 and 4 as therapeutic agents in stroke and brain hemorrhage, leading to poor prognosis. Their approach entails a new and substantial contribution to current literature on the subject matter; however, I have the following major but serious concerns:

  1. This review study is largely confirmatory of a previously published study by Mediators Inflamm. 2010;2010:704202.: Mediators Inflamm. 2013;2013:124614.; J Neuroinflammation. 2013 Feb 17;10:27., Cell Mol Life Sci. 2019 Feb;76(3):523-537. Preclinical studies underscore the TLR2 & TLR4 signaling belongs to a large family of pattern recognition receptors that play a key role in innate immunity and inflammatory responses have been implicated as important biomarkers for ICH-induced inflammation and brain injury, and therefore lacks significant novelty of this review.

  1. This review is too long and, in many places repetitive. Authors should review the paper content for redundancy and ensure only essentials are left.

  1. There are also a few errors in English language grammar that require the authors' attention. 

  1. All abbreviations must be defined when they are first used.

  1. Authors may want to provide a more representative Graphical Abstract

  1. This manuscript cannot be accepted in its present form, I recommend REJECTION

Author Response

This systematic review by Josh Lua is a comprehensive and state-of-the-art presentation of the new insight into the potential role of soluble Toll-like receptors 2 and 4 as therapeutic agents in stroke and brain hemorrhage, leading to poor prognosis. Their approach entails a new and substantial contribution to current literature on the subject matter; however, I have the following major but serious concerns:

  1. This review study is largely confirmatory of a previously published study by Mediators Inflamm. 2010;2010:704202.: Mediators Inflamm. 2013;2013:124614.; J Neuroinflammation. 2013 Feb 17;10:27., Cell Mol Life Sci. 2019 Feb;76(3):523-537. Preclinical studies underscore the TLR2 & TLR4 signaling belongs to a large family of pattern recognition receptors that play a key role in innate immunity and inflammatory responses have been implicated as important biomarkers for ICH-induced inflammation and brain injury, and therefore lacks the significant novelty of this review.

REPLY: What is unique here is that our review focuses specifically on the interactions of soluble receptor variants as well as heme. In regards to, Mediators Inflamm. 2010;2010:704202.: Mediators Inflamm. 2013;2013:124614.; and Cell Mol Life Sci. 2019 Feb;76(3):523-537, these papers focus more on the role of ischemia-reperfusion injuries. While this might show that TLRs have therapeutic potential in ischemic stroke, our review mainly covers hemorrhagic stroke. J Neuroinflammation. 2013 Feb 17;10:27. Is a very relevant paper to our study, and we have opted to include it in our paper. However, it should be noted that this paper mentions nothing about metHb or solubles TLRs. Furthermore, the paper was written in 2013, while our current paper focuses on all literature up until the time of writing (2021). We believe that our paper is not only unique, but it also contributes to the body of science as a whole because it summarizes the literature and proposes future directions for research, especially regarding the therapeutic application of sTLR.

  1. This review is too long and, in many places repetitive. Authors should review the paper content for redundancy and ensure only essentials are left.

REPLY: We have now reviewed thoroughly the content and eliminated some of the content which we believed to be repetitive/extraneous information. 

  1. There are also a few errors in English language grammar that require the authors' attention. 

REPLY: We sincerely apologize for any typos. This review has now been edited by an English Native Professional Medical Editor and should be up to the standard of publication now.

  1. All abbreviations must be defined when they are first used.

REPLY: We have reviewed the paper and correctly defined abbreviations as they were used, including the names of proteins like TNFα (line 84) and ADAM10/17 (line 66).

  1. Authors may want to provide a more representative Graphical Abstract

REPLY: We believe that our abstract and figure can accurately represent the content of our review.

  1. This manuscript cannot be accepted in its present form, I recommend REJECTION. 

REPLY: We thank the reviewer for their constructive comments, but we believe that our review has its merits and provides a novel take on the current literature regarding soluble TLRs and heme/hemoglobin. With that being said, we have revised our manuscript based on their feedback.

Reviewer 3 Report

Thank you very much for this interesting review providing pathophysiological insights into the role of TLRs antagonism in hemorrhagic stroke to optimize patient's outcome by diminishing inflammatory processes.

Introduction is very well written, but needs clearly to be shortened.

Methodology is not present in the manuscript. Is it not necessary? If yes I cant find a word about the use of PRISMA guidelines, which is mandatory.

Discussion. Well written, interesting works to be mentioned should be:

-The Role of Losartan as a Potential Neuroregenerative Pharmacological Agent after Aneurysmal Subarachnoid Haemorrhage

-Aspirin treatment prevents inflammation in experimental bifurcation aneurysms in New Zealand White rabbits

Author Response

Thank you very much for this interesting review providing pathophysiological insights into the role of TLRs antagonism in hemorrhagic stroke to optimize patient outcomes by diminishing inflammatory processes.

The introduction is very well written but needs clearly to be shortened.

Methodology is not present in the manuscript. Is it not necessary? If yes I can’t find a word about the use of PRISMA guidelines, which is mandatory.

REPLY: We followed IJMS’ Instructions to Authors, which is why we did not include a methodology section. While we are not following the PRISMA guidelines per se, this review did an exhaustive literature review in Google Scholar, Embase, Pubmed, OneSearch, UF Library, MyNCBI, and Clinicaltrials.gov. We used the following search terms: ("Hemoglobin" OR "heme" OR "oxheme" OR "hemin" OR "oxidized hemoglobin" OR "Hb" OR "methemoglobin" OR "MetHb") AND ("soluble toll like receptor" OR "sTLR" OR "TLR2" OR "TLR4" OR "TLR"), and another search using: (previous terms) AND ("stroke" OR "SAH" OR "ICH" OR "ischemic stroke")*

Discussion. Well written, interesting works to be mentioned should be:

-The Role of Losartan as a Potential Neuroregenerative Pharmacological Agent after Aneurysmal Subarachnoid Haemorrhage

-Aspirin treatment prevents inflammation in experimental bifurcation aneurysms in New Zealand White rabbits

REPLY: While these works are interesting, we are not entirely sure why these references should be added to this manuscript. In regard to Losartan, it prevents the binding of angiotensin and as such plays no role in the mechanisms of Toll-like receptors and heme. In regard to Aspirin, it is a nonselective NSAID and there is no clear documentation that it acts through the pathways being investigated during the review. As mentioned by Reviewer 2, to keep the manuscript more concise, we are focussing on the specific interactions of soluble toll-like receptors and heme, and consequently are not including these 2 references.

Round 2

Reviewer 2 Report

Accept in present form